# Phenol-chloroform-based RNA purification for detection of SARS-CoV-2 by RT-qPCR: Comparison with automated systems

**Henrik Dimke**[1,2], **Sanne L. Larsen**[3], **Marianne N. Skov**[3], **Hanne Larsen**[3], **Gitte N. Hartmeyer**[3], **Jesper B. Moeller**[4,5]*

**1** Department of Cardiovascular and Renal Research, Institute of Molecular Medicine, University of Southern Denmark, Odense, Denmark, **2** Department of Nephrology, Odense University Hospital, Odense, Denmark, **3** Department of Clinical Microbiology, Odense University Hospital, Odense, Denmark, **4** Department of Cancer and Inflammation Research, Institute of Molecular Medicine, University of Southern Denmark, Odense, Denmark, **5** Danish Institute for Advanced Study, University of Southern Denmark, Odense, Denmark

* jbmoeller@health.sdu.dk

**Data Availability Statement:** All relevant data are within the paper and its Supporting information files.

## Abstract

The outbreak of severe acute respiratory syndrome coronavirus 2 (SARS-CoV-2) rapidly reached pandemic levels. Sufficient testing for SARS-CoV-2 has remained essential for tracking and containing the virus. SARS-CoV-2 testing capabilities are still limited in many countries. Here, we explore the use of conventional RNA purification as an alternative to automated systems for detection of SARS-CoV-2 by RT-qPCR. 87 clinical swab specimens were extracted by conventional phenol-chloroform RNA purification and compared to commercial platforms for RNA extraction and the fully integrated Cobas®6800 diagnostic system. Our results show that the conventional RNA extraction is fully comparable to modern automated systems regarding analytical sensitivity and specificity with respect to detection of SARS-CoV-2 as evaluated by RT-qPCR. Moreover, the method is easily scalable and implemented in conventional laboratories as a low cost and suitable alternative to automated systems for the detection of SARS-CoV-2.

## Introduction

The severe acute respiratory syndrome coronavirus 2 (SARS-CoV-2) causing the coronavirus disease 2019 (COVID-19) has rapidly reached pandemic levels, with COVID-19 related morbidities and mortalities rising in many countries [1–3]. Adequate testing is necessary for tracking the spread of infections, both in the healthcare system and in the general public. Commercial systems and kits are expensive and not readily available in all countries. In the wake of this, several new methods have been proposed to overcome the bottleneck posed by RNA purification in an effort to detect the novel coronavirus (SARS-CoV-2) and clinically diagnose COVID-19 [4, 5]. As such, the acid guanidinium thiocyanate-phenol-chloroform (AGPC) method of RNA extraction has recently been found suitable for SARS-CoV-2 PCR detection [4]. However, it is unclear how it compares to automated systems currently in use at clinical laboratories and if levels of analytical sensitivity, specificity and accuracy are comparable.

**Funding:** The authors received no specific funding for this work.

**Competing interests:** The authors have declared that no competing interests exist.

The AGPC is a simple method used in many research institutions and laboratories worldwide. The method itself has a long-known track record [6] and commercially mixed reagents needed for this step are readily available from several vendors as e.g. TRIzol™ (Invitrogen) and TRI Reagent® (Merck) or can be produced locally from base chemicals at a low cost [6]. For COVID-19 testing it is currently unknown whether the AGPC-based RNA extraction method can perform at levels similar to automated testing systems. We therefore aimed to compare the AGPC method to automated systems commonly used in clinical detection of SARS-CoV-2 and to evaluate its suitability as a replacement for automated systems when proprietary materials are not readily available.

## Materials and methods

A total of 87 clinical sample specimens were obtained from patient material undergoing routine diagnostic analyses at the Department of Clinical Microbiology, Odense University Hospital. The samples were obtained from oropharyngeal or nasopharyngeal swabs collected in ESwab™ tubes containing 1 ml of liquid Amies medium (Copan) and used in this study to evaluate the analytical sensitivity, specificity and accuracy of our in-house SARS-CoV-2 RT-qPCR assay after RNA purification using the AGPC method or the automated Maxwell® RSC 48 instrument (Promega). The patient specimens were selected based on SARS-CoV-2 status from the routine diagnostic analyses to reflect a broad range of viral titers in the sample material as determined by the *SARS-CoV-2 E* gene (Ct value range 16–38) using the fully integrated Cobas®6800 diagnostic system (Roche).

Phenol-chloroform extraction was carried out according to the instructions of the Tri Reagent® manufacturer (Merck) with minor modifications (as outlined in S1 Protocol). RNA extraction was compared to the Maxwell® RSC Viral Total Nucleic Acid Purification Kit (Promega) using the Maxwell® instrument according to the manufacturer's recommendations, without the initial heat incubation step. SARS-CoV-2 was detected according to the real-time PCR protocol established by Corman et al. [7]. The fully automated IVD-CE-labelled Cobas®6800 system (Roche Diagnostics, Basel, Switzerland) was used in this study (set as gold standard) for the evaluation of Maxwell® and AGPC methods of RNA purification. Testing was performed using the Cobas® SARS-CoV-2 test assay with proprietary primers directed against the *SARS-CoV-2 ORF1* and *E*-gene.

To distinguish between true negative results and reactions affected by inadequate RNA isolation, presence of PCR inhibitors or instrument failure, Nobilis ND C2 vaccine (NDV) against Newcastle Disease (Nobilis) was added to the ESwab™ media as an internal control prior to RNA purification using the AGPC method and the Maxwell® instrument. The amount of added NDV had previously been titrated to yield a Ct value ~27.5 as measured by RT-qPCR and routinely used for evaluating sample quality after RNA isolation at the Department of Clinical Microbiology. To determine a Ct cutoff value for NDV after purification with the Maxwell® instrument and the AGPC method, 24 clinical swab samples with known SARS CoV-2 status based on the Cobas®6800 system were processed using both methods and the data for the in-house NDV and SARS-CoV-2 RT-qPCR assays were evaluated. Based on these data and applying a precautionary principle only samples yielding Ct values below 29.5 for the internal control NDV assessed by RT-qPCR on the Maxwell® and TRI Reagent® platforms were included in the analyses.

### Statistical analyses

All statistical tests were performed with Prism v8.3 (GraphPad). The sensitivity, specificity and accuracy of our in-house SARS-CoV-2 RT-qPCR assay after RNA purification using the

Maxwell® RSC 48 and AGPC method were evaluated by comparison to results obtained for the SARS-CoV-2 *E* gene using the Cobas®6800 diagnostic system. Sensitivity is defined as the probability of a test result being positive for a SARS-CoV-2 positive sample. Specificity is defined as the probability of a test result being negative for a SARS-CoV-2 negative sample. Accuracy is defined as the probability of a sample being correctly diagnosed. 95% Confidence intervals were calculated using the "exact" Clopper-Pearson method. Pearson correlation coefficients (r) were calculated for Ct correlation studies. Paired Student's t test was applied for analyses of paired samples with one independent variable.

### Ethics statement

Exception from review by the ethical committee system and informed consent was given by the Regional Committees on Health Research Ethics for Southern Denmark in agreement with Danish law on assay development projects.

## Results

To evaluate whether conventional AGPC based extraction of RNA could serve as a viable alternative to automated systems with respect to reliability and accuracy, we isolated RNA using the AGPC method from 87 clinical specimens (oropharyngeal or nasopharyngeal swabs) with known SARS-CoV-2 status (57 positive and 30 negative), and performed a side-by-side comparison with the identical samples extracted on a Maxwell® instrument. The samples used had previously been analyzed using the state-of-the-art fully integrated Cobas®6800 diagnostic system capable of analyzing patient specimens from sample to test result without manual interference, using proprietary primers directed against the *SARS-CoV-2 ORF1* and *E*-gene. Analysis of the sample specimens with RNA extracted by the Maxwell® instrument showed a 98.2% sensitivity, 96.4% specificity and 97.6% accuracy, but also reported a single false positive and a false negative sample compared to the Cobas®6800 system (Fig 1A). Importantly, analysis of the sample specimens using the AGPC method for RNA extraction displayed a 98.0% sensitivity, 100% specificity and 98.8% accuracy, with no false positive and only 1 false negative compared to Cobas®6800 system.

To further validate our findings, we performed a direct comparison of the Ct values measured for the *SARS-CoV-2 E* gene in the SARS-CoV-2 positive sample specimens after AGPC extraction of the RNA, to the Ct values from identical samples reported using the Cobas®6800 system and the Maxwell® instrument (Fig 1B and 1C), respectively. The Ct values attained for the *SARS-CoV-2 E* gene using the AGPC method showed a highly significant correlation to those reported for the Cobas®6800 system (r = 0.97, p<0.0001) and the Maxwell® instrument (r = 0.98, p<0.0001). Thus, to assess the purity, yield and efficiency of the RNA extraction using the AGPC method we compared the Ct values reported for the NDV to those attained for the same identical samples extracted using the Maxwell® instrument. Direct comparison of the Ct values revealed no significant differences in Ct values for NDV between the AGPC method and the Maxwell® RSC 48 instrument (Fig 1D).

Using standard laboratory equipment, we have adapted and modified the standard AGPC protocol for use in a large-scale RNA extraction pipeline (S1 Protocol).

## Discussion

We find that the AGPC method is a reliable method for RNA extraction fully comparable to automated systems with respect to detection of SARS-COV-2 in oropharyngeal and nasopharyngeal samples isolated from patients with suspected cases of COVID-19. The AGPC method of RNA extraction relies on simple chemical mixtures that can be easily obtained from

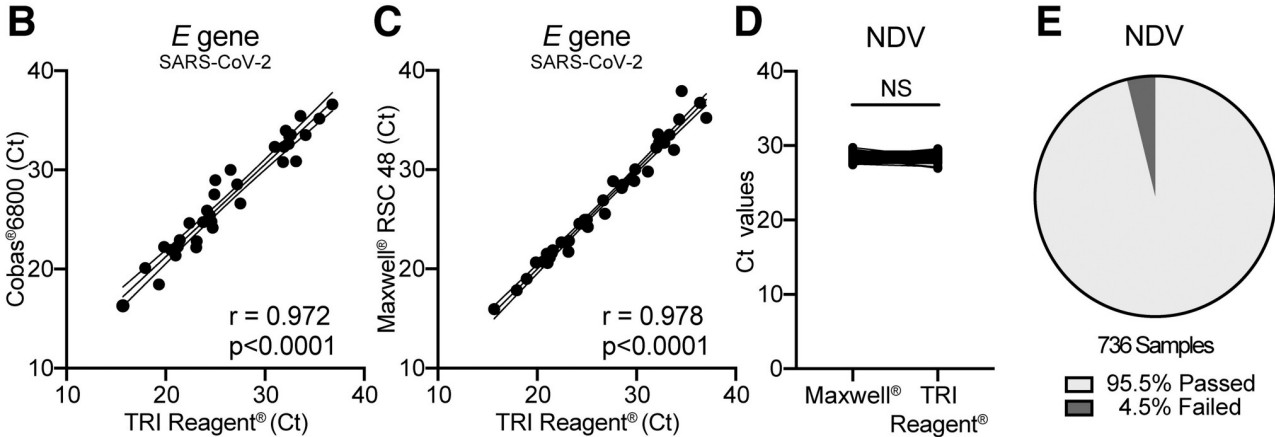

**A**

| Method | TP | TN | FP | FN | Sensitivity (95% CI) | Specificity (95% CI) | Accuracy (95% CI) |
|---|---|---|---|---|---|---|---|
| Cobas®6800 System (Roche) | 57* | 30* | 0* | 0* | NA | NA | NA |
| Maxwell® RSC 48 (Promega) | 53 | 27 | 1 | 1 | 98.2% (90.1-100) | 96.4% (81.7-99.9) | 97.6% (91.5-99.6) |
| AGPC method (Tri Reagent®) | 50 | 29 | 0 | 1 | **98.0%** (89.6-100) | **100%** (90.2-100) | **98.8%** (93.2-100) |

*SARS-CoV-2 status based on the initial clinical analysis using the Cobas®6800 system.*
True positive (TP), true negative (TN), false positive (FP), false negative (FN) and confidence interval (CI)

**Fig 1. Phenol-chloroform extraction of RNA is a viable alternative to automated systems, showing similar levels of sensitivity, specificity and accuracy. (A)** 87 patient specimens with known SARS-CoV-2 status based upon routine testing for SARS-CoV-2 using the Cobas®6800 platform were compared to results reported for the in-house SARS-CoV-2 RT-qPCR assay after RNA isolation using the Maxwell® RSC 48 instrument or TRI Reagent®. Only RNA specimens passing the internal control for the Newcastle Disease Virus vaccine strain (NDV, Ct values <29.5) were used for the analyses (Maxwell®; 82 samples, TRI Reagent®; 80 samples). True positive (TP), true negative (TN), false positive (FP), false negative (FN), confidence interval (CI). Sensitivity is defined as the probability a test result is positive for a SARS-CoV-2 positive sample. Specificity is defined as the probability a test result is negative for a SARS-CoV-2 negative sample. Accuracy is defined as the probability a patient sample is correctly evaluated for SARS-CoV-2. **(B)** Diagram showing a highly significant correlation (r = 0.970, p<0.0001) between obtained Ct values for the SARS-CoV-2 $E$ gene in SARS-CoV-2 positive specimens when assessed by RT-qPCR using the Cobas®6800 platform and the in-house SARS-CoV-2 RT-qPCR assay after RNA isolation using TRI Reagent®. **(C)** Diagram showing a highly significant correlation (r = 0.978, p<0.0001) between obtained Ct values for the SARS-CoV-2 $E$ gene in SARS-CoV-2 positive specimens when assessed by RT-qPCR using the in-house RT-qPCR assay after RNA isolation using the Maxwell® RSC 48 instrument and TRI Reagent®. **(D)** Side-by-side comparison of Ct values obtained for the internal control NDV when assessed by RT-qPCR using identical patient specimens from which RNA was isolated using the Maxwell® RSC 48 instrument or the AGPC method. Not significant (NS). **(E)** Pie chart showing the average rate of sample specimens that pass the NDV internal control for RNA extraction and sample quality after isolation of the RNA using the AGPC pipeline (n = 736).

commercial vendors or mixed by combining the needed chemicals. Overall, the method is low-tech and routinely used in the many laboratories worldwide. The use for the AGPC method as a substitute for automated systems is especially important when automated systems are readily available but strained due to supply shortages. Furthermore, in places where these automated systems are not available, the AGPC method may allow for detection of viruses to levels comparable to those of the modern automated systems.

The AGPC method is robust and near the level of advanced commercial methods. When compared to the Maxwell®-based automation of RNA extraction using the same individual samples and the same SARS-COV-2 real-time PCR assay run in parallel, the results are virtually identical. For the Cobas®6800 system, the Ct values reported for the SARS-CoV-2 $E$ gene

were similarly comparable to those reported for the AGPC method. However, the Cobas®6800 system requires twice the volume of sample specimen for isolation of the RNA, utilizes approximately four times the amount of RNA product for the RT-qPCR analysis and is based upon proprietary primers and probes for detection of the SARS-CoV-2 *E* gene. Thus, a direct comparison cannot be made between the Cobas®6800 system and the results obtained with the used in-house SARS-COV-2 PCR assay.

The state-of-the-art Cobas®6800 system has a capacity of 384 samples per 8 hours, which is roughly equivalent to the throughput of our RNA isolation pipeline presented here. However, the system is unsurpassed in ease, accuracy and sensitivity. At the onset and progression of a pandemic where easy and reliable screening is critical; these fully integrated systems can reduce the time needed for identification of virus-positive cases, which is critical to limit the spread. We initiated this comparative study during the initial phase of the current COVID-19 pandemic because there was a worldwide shortage of kits and reagents for automated systems, which underlines the importance of redundant methods that can be applied at a low cost, independent of proprietary reagents and commercial interest.

Here we show that the AGPC method is easily scalable to volumes usable for clinical diagnostics (S1 Protocol). As with any work with viruses there is a chance of viral contamination. However, the collected sample specimens are mixed directly in organic solvent, which efficiently inactivates coronavirus. Hence, only this initial step needs to be carried out in a specialized class II facility [4, 8, 9].

## Supporting information

**S1 Protocol.**
(PDF)

## Acknowledgments

This manuscript has been released as a pre-print at MedRxiv.org, Dimke et al. [10]. We thank the technicians at the Institute for Molecular Medicine, University of Southern Denmark and the Department of Clinical Microbiology, Odense University Hospital for their assistance during testing, validation and implementation of the AGPC method to expand SARS-CoV-2 testing capacity. We also thank dr. Kurt J. Handberg, Department of Clinical Microbiology, Skejby, Denmark, for kindly sharing primers and probe sequences for the detection of NDV.

## Author Contributions

**Conceptualization:** Henrik Dimke, Gitte N. Hartmeyer, Jesper B. Moeller.

**Data curation:** Jesper B. Moeller.

**Formal analysis:** Henrik Dimke, Sanne L. Larsen, Jesper B. Moeller.

**Investigation:** Henrik Dimke, Sanne L. Larsen, Hanne Larsen, Jesper B. Moeller.

**Methodology:** Henrik Dimke, Sanne L. Larsen, Marianne N. Skov, Gitte N. Hartmeyer, Jesper B. Moeller.

**Resources:** Marianne N. Skov.

**Supervision:** Marianne N. Skov, Gitte N. Hartmeyer.

**Validation:** Hanne Larsen, Jesper B. Moeller.

**Visualization:** Jesper B. Moeller.

**Writing – original draft:** Henrik Dimke, Jesper B. Moeller.

**Writing – review & editing:** Henrik Dimke, Sanne L. Larsen, Marianne N. Skov, Hanne Larsen, Gitte N. Hartmeyer, Jesper B. Moeller.

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
