## [Decision Letter · Decision Letter 0]

14 Jan 2021

PONE-D-20-39657

Phenol-chloroform-based RNA purification for detection of SARS-CoV-2 by RT-qPCR: comparison with automated systems

PLOS ONE

Dear Dr. Moeller,

Thank you for submitting your manuscript to PLOS ONE. After careful consideration, we feel that it has merit but does not fully meet PLOS ONE’s publication criteria as it currently stands. Therefore, we invite you to submit a revised version of the manuscript that addresses the points raised during the review process.

The reviewers find this study well presented and informative.

My personal opinion is that Phenol-chloroform-based RNA purification is, in principle, already in the past. For diagnostic purposes, isolation of total RNA or DNA can be avoided.

A few comments from reviewer #2 that must be taken into account when preparing the manuscript.

We look forward to receiving your revised manuscript.

Kind regards,

Ruslan Kalendar, PhD

Academic Editor

PLOS ONE

Journal Requirements:

2. To meet PLOS ONE submission guidelines, in your Methods section, please provide additional information regarding your statistical analyses. For more information on PLOS ONE's expectations for statistical reporting, please see https://journals.plos.org/plosone/s/submission-guidelines.#loc-statistical-reporting.

Reviewers' comments:

Reviewer's Responses to Questions

**Comments to the Author**

1. Is the manuscript technically sound, and do the data support the conclusions?

Reviewer #1: Yes

Reviewer #2: Yes

Reviewer #3: Yes

2. Has the statistical analysis been performed appropriately and rigorously? 

Reviewer #1: Yes

Reviewer #2: Yes

Reviewer #3: Yes

3. Have the authors made all data underlying the findings in their manuscript fully available?

Reviewer #1: Yes

Reviewer #2: Yes

Reviewer #3: Yes

4. Is the manuscript presented in an intelligible fashion and written in standard English?

Reviewer #1: Yes

Reviewer #2: Yes

Reviewer #3: Yes

5. Review Comments to the Author

Reviewer #1: This is a focused and straightforward examination of the efficacy of the guanidinium thiocyanate-phenol-chloroform (AGPC) method of RNA extraction for RNA sample preparation used in the detection of SARS-CoV-2 by RT-qPCR. The primary objective of the authors was to determine whether the much less expensive and technologically simpler, although more labor intensive, bench top method of RNA isolation from nasopharyngeal and oropharyngeal swabs is as effective as automated RNA isolations carried out by two different instruments currently in use for COVID-19 testing. The rational for this study is that evidence supporting efficacy of the AGPC extraction method can enable COVID-19 testing to be carried out in locations that lack the resources for expensive, high-throughput RNA isolation devices. The results clearly show that the bench top method can reliably be used in place of automated instruments.

The study is technically sound and well controlled, and the data is appropriately analyzed for statistical significance. The experiments and results are thoroughly and clearly described in well-written, grammatically correct English.

**Reviewer #2: **

The manuscript entitled "Phenol-chloroform-based RNA purification for detection of SARS-CcV-2 by RT qPCR: comparison with automated systems" is a valuable contribution to the field of SARS-CoV-2 detection systems.

However, although conventional RNA purification can be a valid alternative to commercial platforms for RNA extraction, it is also true that in a pandemic setting when every moment matters, the use of commercial kits and automated system for the detection of SARS-COv-2 can speed up the identification process of SARS-CoV-2 positive subjects. Maybe the Authors should discuss this point better in their manuscript.

**line 135: in the Materials and Methods section, the Authors indicated that both oropharyngeal and nasopharyngeal swabs were collected. Please make the required correction.**

Reviewer #3:

 In this paper by Moeller lab, they compare the result of the standard phenol choloform isolation of SARS-CoV-2 RNA to automated RNA extraction systems. They use qRT-PCR to detect the inactivated viral RNA and compare the two methods head to head. Appropriate positive controls are used. They find that the PC method is comparable to the automated systems in terms of true positive (TP) TN FP FN. Importantly a detailed protocol is presented in the supplemental data. While this method is typically successfully performed by those skilled in working with RNA and therefore more human error could be introduced during the isolation, it does offer an alternative to automated systems in developing countries that may not have access to those equipment. While PC extraction of SARS-CoV-2 RNA has been preciously reported (their ref 5) this study for the first times does a head to head comparison of the two methods and shows them to be equivalent.

---

## [Author Response · Author response to Decision Letter 0]

26 Jan 2021

Response to Editor and Reviewer’s comments

Academic Editor:

• We have formatted and rearranged the manuscript to meet all PLOS ONE’s requirements.

2. To meet PLOS ONE submission guidelines, in your Methods section, please provide additional information regarding your statistical analyses. 

• We have added additional information regarding statistical analyses used to the Methods section. These changes can be found on line 119-129.

• We have moved the Ethics statement to the Methods section of the manuscript. These changes can be found on line 131-134.

Reviewer #1:

This is a focused and straightforward examination of the efficacy of the guanidinium thiocyanate-phenol-chloroform (AGPC) method of RNA extraction for RNA sample preparation used in the detection of SARS-CoV-2 by RT-qPCR. The primary objective of the authors was to determine whether the much less expensive and technologically simpler, although more labor intensive, bench top method of RNA isolation from nasopharyngeal and oropharyngeal swabs is as effective as automated RNA isolations carried out by two different instruments currently in use for COVID-19 testing. The rational for this study is that evidence supporting efficacy of the AGPC extraction method can enable COVID-19 testing to be carried out in locations that lack the resources for expensive, high-throughput RNA isolation devices. The results clearly show that the bench top method can reliably be used in place of automated instruments.

The study is technically sound and well controlled, and the data is appropriately analyzed for statistical significance. The experiments and results are thoroughly and clearly described in well-written, grammatically correct English.

• We thank the Reviewer for their kind review.

Reviewer #2:

The manuscript entitled "Phenol-chloroform-based RNA purification for detection of SARS-CcV-2 by RT qPCR: comparison with automated systems" is a valuable contribution to the field of SARS-CoV-2 detection systems.

However, although conventional RNA purification can be a valid alternative to commercial platforms for RNA extraction, it is also true that in a pandemic setting when every moment matters, the use of commercial kits and automated system for the detection of SARS-COv-2 can speed up the identification process of SARS-CoV-2 positive subjects. Maybe the Authors should discuss this point better in their manuscript.

• We appreciate and agree with the Reviewer’s suggestion. We have included a short discussion of the benefits of automated systems for rapid identification of SARS-CoV-2 positive subjects during a pandemic. These additions can be found on line 205-207 and line 222-255.

Line 135: in the Materials and Methods section, the Authors indicated that both oropharyngeal and nasopharyngeal swabs were collected. Please make the required correction.

• We have corrected this error. The addition can be found on line 198-199

Reviewer #3:

In this paper by Moeller lab, they compare the result of the standard phenol choloform isolation of SARS-CoV-2 RNA to automated RNA extraction systems. They use qRT-PCR to detect the inactivated viral RNA and compare the two methods head to head. Appropriate positive controls are used. They find that the PC method is comparable to the automated systems in terms of true positive (TP) TN FP FN. Importantly a detailed protocol is presented in the supplemental data. While this method is typically successfully performed by those skilled in working with RNA and therefore more human error could be introduced during the isolation, it does offer an alternative to automated systems in developing countries that may not have access to those equipment. While PC extraction of SARS-CoV-2 RNA has been preciously reported (their ref 5) this study for the first times does a head to head comparison of the two methods and shows them to be equivalent.

• We thank the Reviewer for their kind comments.

---

## [Editor Report · Decision Letter 1]

9 Feb 2021

Phenol-chloroform-based RNA purification for detection of SARS-CoV-2 by RT-qPCR: comparison with automated systems

PONE-D-20-39657R1

Dear Dr. Moeller,

We’re pleased to inform you that your manuscript has been judged scientifically suitable for publication and will be formally accepted for publication once it meets all outstanding technical requirements.

Kind regards,

Ruslan Kalendar, PhD

Academic Editor

PLOS ONE

---

## [Editor Report · Acceptance letter]

15 Feb 2021

PONE-D-20-39657R1 

Phenol-chloroform-based RNA purification for detection of SARS-CoV-2 by RT-qPCR: comparison with automated systems 

Dear Dr. Moeller:

I'm pleased to inform you that your manuscript has been deemed suitable for publication in PLOS ONE. Congratulations! Your manuscript is now with our production department. 

Kind regards, 

on behalf of

Prof. Ruslan Kalendar 

Academic Editor

PLOS ONE